# A Study of the Effects of Transfer Learning on Adversarial Robustness

**Pratik Vaishnavi**  *pvaishnavi@cs.stonybrook.edu*
*Stony Brook University*

**Kevin Eykholt**  *kheykholt@ibm.com*
*IBM Research*

**Amir Rahmati**  *amir@cs.stonybrook.edu*
*Stony Brook University*

**Reviewed on OpenReview:** *https://openreview.net/forum?id=T6RygOFZ6B*

## Abstract

The security and robustness of AI systems are paramount in real-world applications. Previous research has focused on developing methods to train robust networks, assuming the availability of sufficient labeled training data. However, in deployment scenarios with limited training data, existing techniques for training robust networks become impractical. In such low-data scenarios, non-robust training methods often resort to *transfer learning*. This involves pre-training a network on a large, possibly labeled dataset and fine-tuning it for a new task with a limited set of training samples. The efficacy of transfer learning in enhancing adversarial robustness is not comprehensively explored. Specifically, it remains uncertain whether transfer learning can improve adversarial performance in low-data scenarios. Furthermore, the potential benefits of transfer learning for certified robustness are unexplored. In this paper, we conduct an extensive analysis of the impact of transfer learning on both empirical and certified adversarial robustness. Employing supervised and self-supervised pre-training methods and fine-tuning across 12 downstream tasks representing diverse data availability scenarios, we identify the conditions conducive to training adversarially robust models through transfer learning. Our study reveals that the effectiveness of transfer learning in improving adversarial robustness is attributed to an increase in standard accuracy and not the direct "transfer" of robustness from the source to the target task, contrary to previous beliefs. Our findings provide valuable insights for practitioners aiming to deploy robust ML models in their applications. The code used to produce the findings in this paper is available at: https://github.com/Ethos-lab/transfer_learning_for_adversarial_robustness

## 1 Introduction

Transfer learning, as summarized in Figure 1, has been extensively studied for improving standard generalization in machine learning systems across various data availability scenarios (Yosinski et al., 2014; Kornblith et al., 2019; He et al., 2019). In the context of adversarial robustness, however, there are only limited works that have studied the benefits of transfer learning (Hendrycks et al., 2019; Chen et al., 2020a). These works generally limit themselves to empirical robustness by solely using adversarial training (Madry et al., 2018) (or its variants) in their experiments. Furthermore, they only study the scenario where abundant data is available for the downstream tasks, *i.e.*, well-represented tasks (*e.g.*, CIFAR-10, CIFAR-100). The exact effect of transfer learning on empirical robustness when there is a lack of abundant data for the downstream tasks, *i.e.*, under-represented tasks, is therefore unknown.

It is also unclear whether the findings in the context of empirical robustness would apply to certified robustness training methods, specifically randomized smoothing-based methods (Cohen et al., 2019; Salman et al., 2019; Zhai et al., 2020; Jeong & Shin, 2020; Jeong et al., 2021) which provide state-of-the-art certified robustness in the $\ell_2$-space. This is because both these classes of methods rely on fundamentally different ways of measuring and encoding adversarial robustness, and so classifiers trained using them inherit different properties. Case in point, Kireev et al. (2022) demonstrated that empirical and certified training methods exhibit dissimilar levels of robustness against common image corruptions. Finally, there is little work that studies the effect of self-supervised pre-training on adversarial robustness, with existing works limiting themselves to well-represented tasks.

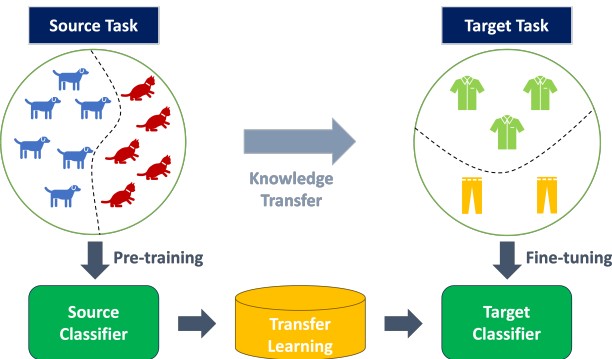

Figure 1: Through transfer learning, one can obtain high-performance networks in settings where it would otherwise be infeasible, *i.e.,* low-data regimes. The network is first trained on a source task with a large training dataset and then fine-tuned on the low-data target task.

Table 1 summarizes the findings of prior works in regard to improving performance and robustness in a range of data availability scenarios. The effects of transfer learning on adversarial robustness are largely unexplored (limited to empirical robustness and well-represented tasks). Furthermore, we note that self-supervised pre-training has become an important component of the transfer learning framework of late as it alleviates the need for labeled data for pre-training. The models fine-tuned using pre-trained weights generated via self-supervised learning have exhibited unprecedented generalization ability, unlocking large-scale commercial applications that were infeasible only a few years back. However, using self-supervision to train highly secure ML models is a topic that has largely been overlooked. Therefore, in this paper, we make adversarial robustness our primary focus and broadly study the effects of self-supervised pre-training on it.

We summarize the **contributions** of this work as follows:

- We perform a comprehensive study on the utility of transfer learning towards certified and empirical robustness across a range of downstream tasks. First, a classifier is robustly pre-trained on a large-scale dataset (*i.e.,* ImageNet) using supervised or self-supervised methods and then robustly fine-tuned on the downstream task. Our experimental results show that such pre-training is beneficial toward improving adversarial performance on downstream tasks compared to training on the downstream task directly.

- We further show that during transfer learning, only the fine-tuning part of the pipeline needs to rely on robust training methods. This finding eases the overhead of training robust classifiers. Also, regardless of the amount of labeled data available for either pre-training or fine-tuning, classifiers with high adversarial robustness can be trained on downstream tasks.

- Finally, our work is the first to demonstrate that classifiers can be trained to achieve high certified robustness on downstream tasks irrespective of the amount of labeled data available, either during pre-training or fine-tuning.

Our findings serve as a useful guidance for ML practitioners wanting to deploy highly robustness models in a range of data availability scenarios.

## 2 Background

In this paper, we focus on transfer learning for image classification tasks. More specifically, we explore whether transfer learning can be used to train deep neural network-based image classifiers with high (empirical and certified) adversarial robustness in a range of data availability scenarios. In this section, we provide

Table 1: Summarizing the findings of prior works regarding the usefulness of transfer learning towards standard generalization and adversarial robustness.

| | | Is Transfer Learning Useful? | | | |
| --- | --- | --- | --- | --- | --- |
| | | Supervised | | Self-Supervised | |
| | | Low-Data | High-Data | Low-Data | High-Data |
| Standard Generalization | | ✔ (Kornblith et al., 2019) | ✘ (He et al., 2019) | ✔ (Chen et al., 2020b) | ✔ (Chen et al., 2020b) |
| Adversarial Robustness | Empirical | ? | ✔ (Hendrycks et al., 2019) | ? | ✔ (Chen et al., 2020a) |
| | Certified | ? | ? | ? | ? |

readers with the necessary background regarding transfer learning (Section 2.1) and adversarial robustness of deep neural networks (Section 2.2).

## 2.1 Transfer Learning

In transfer learning (Caruana, 1994; Pan & Yang, 2009; Bengio et al., 2011; Bengio, 2012; Yosinski et al., 2014; Huh et al., 2016), a network is pre-trained on a source task and then fine-tuned on a target task. Through pre-training, the network learns features that enable it to generalize better when fine-tuned on the target task (Yosinski et al., 2014). This is true even when the source and target tasks are dissimilar. For example, prior works (Sermanet et al., 2013; Girshick et al., 2014) re-purposed networks trained for ImageNet (Deng et al., 2009) classification task to achieve breakthroughs on object detection tasks. Pre-training has also been shown to be an effective solution for training high-performance networks when available training data is insufficient for standard training (Pan & Yang, 2009). However, He *et al.* (He et al., 2019) showed that, in the presence of abundant training data, similar levels of generalization can be achieved whether pre-training is performed or not. In such cases, the only benefit of transfer learning then is faster convergence and, therefore, savings in training time. Other studies found that transfer learning effectively transfers other desirable properties like shape bias (Utrera et al., 2020), robustness to common image corruptions (Yamada & Otani, 2022) and adversarial perturbations (Hendrycks et al., 2019). Specific fine-tuning methods have also been developed to optimally preserve desirable properties like empirical adversarial robustness (Liu et al., 2023; Xu et al., 2023).

### 2.1.1 Self-supervised Pre-training

Traditionally, pre-training was performed in a supervised fashion on large-scale labeled datasets, which can be challenging to acquire in many domains. However, unlabeled data tends to be widely available. To leverage these unlabeled datasets, self-supervised pre-training was proposed to enable models to learn generalizable features by optimizing a custom training objective. Contrastive learning (Chen et al., 2020b; He et al., 2020; Caron et al., 2020; 2021) is one such approach. Models are trained to maximize the similarity between positive pairs (semantically similar data samples) while minimizing the similarity between negative pairs (semantically dissimilar data samples) in the feature space. SimCLR (Chen et al., 2020b), one of the most popular contrastive learning methods, generates the positive pairs by applying two different sets of input transformations (like cropping, color distortion, and blurring) to the same image. Negative pairs are generated using transformed versions of different images. Self-supervised methods often achieve state-of-the-art results in a range of applications such as image classification, object detection, and sentiment analysis after fine-tuning on relatively small amounts of labeled data.

## 2.2 Adversarial Robustness

Neural networks are known to be susceptible to adversarial evasion attacks, which attempt to modify a given input imperceptibly with the goal of triggering misclassification. Since the discovery of this vulnerability,

several methods have been proposed to train neural networks that are robust against such attacks. These methods can be broadly classified as *empirical* and *certified* methods based on the nature of the robustness guarantees they provide.

### 2.2.1 Empirical Adversarial Robustness

Empirical adversarial robustness is traditionally measured using the strongest possible attack within a pre-determined threat model. Robustness training methods that rely on this strategy train the neural network to be robust against this strongest attack and, in turn, gain robustness against all possible attacks within the same threat model. However, such robustness is not provable in nature and can be challenged by an adaptive adversary (Carlini & Wagner, 2017; Athalye et al., 2018; Tramer et al., 2020). Adversarial training (Madry et al., 2018), is one of the most promising empirical robustness methods, as is evident from the fact that the current state-of-the-art methods (Zhang et al., 2019; Wu et al., 2020) is derived from the basic framework proposed by Madry *et al.* (Madry et al., 2018). This framework involves generating adversarial inputs on the fly during training and updating the neural network's weights using them. Furthermore, several works (Tsipras et al., 2019; Ilyas et al., 2019; Augustin et al., 2020) still study the models trained by Madry *et al.* to learn more about adversarial robustness in general. Due to its prominence and in an attempt to fall in line with prior works, we use adversarial training as a representative of empirical robustness training methods.

### 2.2.2 Certified Adversarial Robustness

Despite the progress made towards developing empirical robustness methods with strong robustness guarantees, the lack of provability remains an issue. Provably/certifiably robust training methods remedy this concern by maximizing the lower bound of a neural network's output corresponding to the correct class within a certain range of input perturbations. If, for a given input, the lower bound of the correct class output is higher than the upper bound of all other class outputs, the neural network is provably robust for that input. Computing and maximizing this lower bound for a multi-layer neural network is an NP-hard problem (Katz et al., 2017). In recent literature, several methods have been proposed to approximately compute this lower bound and incorporate it in the training process of the neural network in a scalable manner. Of these, randomized smoothing based methods (Cohen et al., 2019; Salman et al., 2019; Zhai et al., 2020; Jeong & Shin, 2020; Jeong et al., 2021) yield state-of-the-art robustness in the $\ell_2$-space for modern neural networks. Therefore, in this paper, we focus on these methods.

First formalized by Cohen *et al.* (Cohen et al., 2019), randomized smoothing defines the concept of a smooth classifier. Given a base classifier $f_\theta$, the **smooth classifier** $g_\theta$, is defined as follows:

$$g_\theta(x) = \arg\max_{c \in \mathcal{Y}} P_{\eta \sim \mathcal{N}(0, \sigma^2 I)}(f_\theta(x + \eta) = c) \tag{1}$$

Simply put, the smooth classifier returns the class $c$, which has the highest probability mass under the Gaussian distribution $\mathcal{N}(x, \sigma^2 I)$. If, for a given input $x$, the smooth classifier's output $c$ is equal to the ground truth label $y$, it is said to be certifiably robust (with high probability) at $x$. The **certified radius**, *i.e.,* the input radius in which $x$'s prediction is consistent, is given by:

$$CR(g_\theta; x, y) = \frac{\sigma}{2}[\Phi^{-1}(P_\eta(f_\theta(x + \eta) = y)) - \\ \Phi^{-1}(\max_{y' \neq y} P_\eta(f_\theta(x + \eta) = y'))] \tag{2}$$

Randomized smoothing-based robustness training methods focus on maximizing the average certified radius for a given dataset (Cohen et al., 2019; Salman et al., 2019; Zhai et al., 2020; Jeong et al., 2021). Cohen *et al.* (Cohen et al., 2019) simply augmented the training data with Gaussian noise when training the base classifier. Salman *et al.* (Salman et al., 2019) modified the adversarial training objective to work in this new framework. Zhai *et al.* (Zhai et al., 2020) derived a differentiable approximation of the certified radius and directly maximized it during training. Jeong *et al.* (Jeong & Shin, 2020) find that the certified robustness of a smooth classifier can be greatly improved by enforcing the base classifier's outputs over several noisy

copies of a given input to be consistent. They achieve this consistency by using a regularization loss that forces the output for a noisy copy of the input to be closer to the expected output over several noisy copies. Finally, Jeong *et al.* (Jeong et al., 2021) identified that the certified radius of the smooth classifier is aligned with its prediction confidence and used a combination of adversarial training and *mixup* (Zhang et al., 2018) to favorably calibrate the prediction confidence.

## 3 Empirical is not the same as Certified

Broadly, the process of training adversarially robust classifiers can be dissected into two key stages: (i) quantifying the adversarial risk across the training data distribution and (ii) minimizing this adversarial risk during the training process. Empirical methods, such as Adversarial Training, measure the adversarial risk by determining the maximum loss achievable for an input subjected to adversarial manipulations. This entails employing the most potent attack within a predefined threat model. On the other hand, certified methods utilizing randomized smoothing measure adversarial risk by measuring the largest possible radius around a given input within which the classifier's output remains consistent (Equation 2). The fundamental disparity in how adversarial risk is assessed encourages empirical and certified classifiers to manifest distinct robustness properties that the other may not possess. We perform a brief investigation using Gaussian noise image corruption that illustrates the differences between these two robustness classes. Through this demonstration, we underscore the caution required when extending findings from empirical robustness to the realm of certified robustness.

First, we train two ResNet-50 classifiers on ImageNet: (i) using an empirical robustness training method, *i.e.,* Adversarial Training (AT) (Madry et al., 2018), and (ii) using a certified robustness training method, *i.e.,* Consistency Regularization (CR) (Jeong & Shin, 2020). For CR, we use a Gaussian noise distribution with standard deviation $\sigma = 0.5$. Subsequently, we assess the performance of these classifiers on the test set, perturbed by Gaussian noise using different values of $\sigma$. The outcomes are detailed in Table 2. We observe that the AT classifier's performance declines as the value of $\sigma$ increases, whereas the CR classifier performs well only when the $\sigma$ value for the test data is exactly the same as the value used during training. This

Table 2: Performance comparison of a classifier trained using AT (Madry et al., 2018) and a base classifier trained using CR (Jeong & Shin, 2020), under varying levels of Gaussian noise. The AT classifier shows a gradual decline in performance as the noise severity increases, whereas the CR classifier overfits to the level of noise encountered during training (*i.e.,* $\sigma = 0.5$).

| Method | Noise Stddev ($\sigma$) | | | |
|---|---|---|---|---|
| | 0.001 | 0.01 | 0.1 | 0.5 |
| Adversarial Training | 54.5 | 54.5 | 28.5 | 0.1 |
| Consistency Regularization | 19.0 | 19.1 | 21.8 | 60.5 |

phenomenon aligns with observations made by Kireev et al. (2022), indicating that employing Gaussian data augmentation during training, such as in CR, results in the classifier overfitting to the noise at the value of $\sigma$ used during training. Classifiers trained using AT do not exhibit this overfitting behavior.

Next, we evaluate the certified test accuracy of the smooth classifier derived from the aforementioned base classifiers, presenting the results in Table 3. Certified accuracy at a given $\ell_2$ radius, denoted as $r$, represents the proportion of test samples with a certified radius (as per Equation 2) greater than $r$. Recall that a smooth classifier makes predictions through majority voting over outputs from multiple noisy copies of a given input (see Equation 1). Consequently, selecting an appropriate value for the noise parameter $\sigma$ for the smoothing process, one that the base classifier can "handle", is crucial. We make this selection informed by the findings presented in Table 2. The CR base classifier is smoothed using $\sigma = 0.5$, consistent with the value

Table 3: Certified accuracy at different $\ell_2$ radii of smooth classifiers that use base classifiers trained with AT (Madry et al., 2018) and CR (Jeong & Shin, 2020). The smoothing is performed using $\sigma = 0.01$ and 0.5, respectively. AT smooth classifier exhibits no meaningful certified robustness.

| Method | $\ell_2$ radius | | | |
|---|---|---|---|---|
| | 0.0 | 0.5 | 1.0 | 1.5 |
| Adversarial Training | 49.6 | 0.0 | 0.0 | 0.0 |
| Consistency Regularization | 54.8 | 50.1 | 43.8 | 33.5 |

used during training, aligning with established prac-

tices in randomized smoothing literature (Cohen et al., 2019). In the case of the AT base classifier, we opt for $\sigma = 0.01$ since the base classifier demonstrates sharp drop in performance at higher $\sigma$ values. Once again, we observe noteworthy variations in performance depending on the training method of the base classifier. The CR smooth classifier reports non-zero certified test accuracy at various radii, whereas the AT smooth classifier exhibits no certified robustness, i.e., the certified accuracy for any $\ell_2$ radius $r > 0$ is negligible.

Empirical methods, such as AT, have been extensively scrutinized in the literature, with previous studies (Tsipras et al., 2019; Moosavi-Dezfooli et al., 2019; Ilyas et al., 2019; Kireev et al., 2022) unveiling various distinctive properties introduced by these methods. Given the close conceptual connection between certified robustness and empirical robustness, there might be a tendency to extrapolate findings in the context of empirical robustness without verification. However, the results presented in this section emphasize that these two classes of methods exhibit more dissimilarity than one might presume, and straightforwardly translating findings between them may lead to erroneous assumptions. Therefore, as part of our contributions through this study, we diligently validate the conclusions drawn in the realm of empirical robustness and transfer learning to certified robustness (Section 4.3).

## 4 Transfer Learning for Adversarially Robust ML

Commercial systems are becoming increasingly reliant on AI. However, adversarial attacks remain an ever-present issue when considering the trustworthiness of these systems. Unfortunately, training models with high adversarial robustness using current methods requires access to large amounts of labeled data (Schmidt et al., 2018), which is hard to achieve in many deployment scenarios, even in the actively studied image domain. Except for public datasets such as ImageNet, most vision tasks may only have a handful of labeled data samples for training.

In non-robust scenarios, transfer learning is one solution to alleviate the need for abundant training data for a given task. It involves pre-training on a data-rich (source) task followed by fine-tuning on the low-data downstream task to achieve state-of-the-art performance. Unfortunately, the relationship between transfer learning and adversarial robustness has only been studied in one specific scenario, when the downstream task has abundant labeled training samples and *empirical* adversarial robustness is the property of interest. To our knowledge, there are no works that explore using transfer learning to enable the deployment of empirically robust models on small-scale datasets. Furthermore, there are no works that study the relationship between transfer learning and *certified* adversarial robustness.

We present the first comprehensive study on the utility of transfer learning towards adversarial robustness. In Section 4.1, we describe our experiment setup. In Sections 4.2 and 4.3, we examine the benefits of transfer learning in the context of empirical and certified robustness in a range of data availability scenarios. Here, we use different pre-training methods (robust and non-robust), and perform fine-tuning robustly. In Section 5, we will examine the need for robustness training during the different phases of transfer learning, *i.e.,* pre-training and fine-tuning.

### 4.1 Setup

In this section, we describe our experimental setup. Additional implementation details are available in Appendix A.

***Dataset and Model.*** For pre-training (supervised and self-supervised), we use the standard ImageNet dataset. For fine-tuning, we use a suite of 12 downstream datasets (Kornblith et al., 2019) often used in transfer learning literature. Training is done using a ResNet-50 classifier. All images are scaled to $224 \times 224$ in order to be compatible with ImageNet pre-trained weights.

***Threat Model.*** We measure the adversarial robustness with respect to a white-box $\ell_2$ adversarial attack. Our choice of adversary is motivated by the fact that randomized smoothing (Cohen et al., 2019), our choice of certified robustness method, defines robustness in the $\ell_2$ space. This enables us to easily compare both adversarial metrics (empirical and certified) during evaluation.

***Supervised Training.*** As a baseline for comparison, for every downstream task, we train a randomly initialized model using only the downstream task's labeled data. When studying the effects of transfer learning on empirical robustness, we use Adversarial Training (**AT**) (Madry et al., 2018) for baseline training, pre-training, and fine-tuning. AT uses a PGD attack with $\epsilon = 0.5$, step size $= 2\epsilon/3$, and 3 steps. We note that higher values of $\epsilon$ will only result in reducing the overall performance of the models. When studying the effects of transfer learning on certified robustness, we use Consistency Regularization (**CR**) (Jeong & Shin, 2020) for baseline training, pre-training, and fine-tuning. For CR, we use $\sigma = 0.5$, number of Gaussian noise samples $m = 2$, $\lambda = 5$, and $\eta = 0.5$.

***Self-supervised Training.*** Due to its popularity in current literature, we study the benefits of self-supervised pre-training on adversarial robustness. Unfortunately, most existing adversarially robust self-supervised methods (Jiang et al., 2020; Fan et al., 2021; Luo et al., 2022; Xu et al., 2024) have not been scaled to ImageNet, but on smaller datasets instead. The one method we found that uses ImageNet (Gowal et al., 2020) does not have code publicly available. Thus, we use the SimCLR (Chen et al., 2020b) training method, a contrastive learning approach.

***Evaluation.*** For measuring empirical robustness during evaluation, we use autoPGD (Croce & Hein, 2020) (white-box) and Square attack (Andriushchenko et al., 2020) (black-box). We generate adversarial test sets using these two attacks and measure the model's accuracy on them, denoted by **RA-WB** and **RA-BB** respectively. In both cases, we use the same adversarial budget as what was used during training, *i.e.,* $\epsilon = 0.5$. The other attack hyperparameters are set to the default values reported by Croce & Hein (2020), *i.e.,* autoPGD uses 100 steps and 5 random restarts and the Square attack uses 5000 queries and 1 random restart. For measuring certified robustness during evaluation, we use the certification process proposed by Cohen et al. (2019) and report the fraction of inputs in the test set with certified radius (Equation 2) greater than or equal to the adversarial budget of $\epsilon = 0.5$, called certified robust accuracy and denoted as **RA-CT**. Additionally, we report the average radius around an test input within which the model's prediction remains consistent, called Average Certified Radius and denoted as **ACR**. For all models, we also report the accuracy on the clean test set, called standard accuracy and denoted as **SA**.

## 4.2 Empirical Adversarial Robustness

Prior works (Hendrycks et al., 2019; Chen et al., 2020a) have demonstrated that, unlike with standard generalization, empirical robustness benefits from transfer learning for well-represented downstream tasks. We begin our study by validating their findings and then extending them to a wider range of data availability scenarios. On a suite of 12 target tasks, we train three versions of a ResNet-50 classifier: (i) using randomly initialized weights, (ii) using pre-trained weights obtained by performing Adversarial Training (AT) (Madry et al., 2018) on ImageNet, and (iii) using pre-trained weights obtained by performing SimCLR (Chen et al., 2020b) on ImageNet. The standard accuracy (SA) and robust accuracy against white-box and black-box attacks (RA-WB and RA-BB) of the resultant classifiers are reported in Table 4.

First, we see that, as prior work also demonstrated (Hendrycks et al., 2019; Chen et al., 2020a), transfer learning using a model pre-trained using AT improves performance (SA) and robustness (RA-WB and RA-BB) on well-represented downstream tasks (*i.e.,* CIFAR-10, CIFAR-100, and Food). However, our experiments also show that pre-training with AT improves performance and robustness even on under-represented downstream tasks (*e.g.,* Flowers, Pets, and Caltech-101). On average, across all tasks, pre-training with AT improves SA, RA-WB, and RA-BB relative to random initialization by 11.4%, 12.6%, and 13.0% respectively. We also note that SimCLR pre-training yields consistent improvements in SA, RA-WB, and RA-BB, averaging 14.1%, 9.9%, and 15.5% respectively. While improvements in SA were expected, the improvements in RA-WB and RA-BB are surprising given that SimCLR, unlike other self-supervised methods we surveyed (Jiang et al., 2020; Fan et al., 2021; Luo et al., 2022), does not specifically design its objective function with adversarial robustness in mind.

We suspect that improvements in RA due to transfer learning are largely due to the overall improvement in SA rather than the robustness being "transferred" from the source task (ImageNet) to the target tasks. On Birdsnap, for example, both pre-training methods result in lower SA, which is mirrored by lower RA compared to random initialization. In Figure 2, we plot the relative increase in RA-WB vs. the relative

Table 4: Evaluating the benefits of pre-training for empirical adversarial robustness. Given a target task, we train three ResNet-50 classifiers: one using random weight initialization and two using weights pre-trained on a source task (ImageNet). Pre-training is performed using supervised (adversarial training) and self-supervised (SimCLR) objectives. During fine-tuning, the full network is trained using AT. Pre-training improves empirical adversarial robustness across all target tasks. Relative change due to pre-training is denoted in superscript.

| Target Task | Random Init. | | | Sup. Pre-Training | | | Self-Sup. Pre-Training | | |
|---|---|---|---|---|---|---|---|---|---|
| | SA | RA-WB | RA-BB | SA | RA-WB | RA-BB | SA | RA-WB | RA-BB |
| Food | 74.5 | 62.3 | 69.5 | $81.6^{+09.5}$ | $69.2^{+11.1}$ | $77.7^{+11.9}$ | $82.2^{+10.3}$ | $68.6^{+10.0}$ | $78.3^{+12.8}$ |
| CIFAR-100 | 71.8 | 62.5 | 67.4 | $80.1^{+11.6}$ | $70.6^{+12.9}$ | $75.1^{+11.5}$ | $80.9^{+12.6}$ | $70.3^{+12.4}$ | $75.8^{+12.5}$ |
| CIFAR-10 | 93.3 | 88.8 | 90.2 | $95.8^{+02.7}$ | $91.7^{+03.2}$ | $94.0^{+04.2}$ | $95.9^{+02.8}$ | $91.2^{+02.7}$ | $93.8^{+03.9}$ |
| Birdsnap | 65.2 | 50.8 | 59.7 | $61.8^{-05.2}$ | $48.3^{-05.0}$ | $57.1^{-04.3}$ | $60.4^{-07.4}$ | $44.4^{-12.6}$ | $55.5^{-07.1}$ |
| SUN397 | 51.0 | 41.7 | 46.5 | $55.5^{+08.8}$ | $44.4^{+06.5}$ | $52.0^{+11.7}$ | $59.0^{+15.7}$ | $44.3^{+06.3}$ | $55.7^{+19.7}$ |
| Caltech-256 | 61.4 | 54.4 | 58.0 | $70.6^{+14.9}$ | $62.5^{+15.0}$ | $67.4^{+16.3}$ | $76.8^{+25.0}$ | $65.4^{+20.1}$ | $73.4^{+26.6}$ |
| Cars | 88.3 | 83.0 | 86.9 | $87.9^{-00.5}$ | $82.2^{-00.9}$ | $86.4^{-00.6}$ | $85.8^{-02.8}$ | $76.1^{-08.4}$ | $83.9^{-03.5}$ |
| Aircraft | 76.4 | 68.6 | 71.6 | $77.9^{+02.0}$ | $69.6^{+01.5}$ | $74.8^{+04.5}$ | $76.3^{-00.1}$ | $64.6^{-05.8}$ | $74.3^{+03.7}$ |
| DTD | 54.3 | 48.1 | 52.4 | $65.8^{+21.2}$ | $59.7^{+24.1}$ | $64.0^{+22.1}$ | $72.6^{+33.7}$ | $58.9^{+22.4}$ | $70.3^{+34.1}$ |
| Pets | 73.2 | 63.3 | 70.0 | $86.9^{+18.7}$ | $78.4^{+23.8}$ | $83.9^{+19.8}$ | $88.6^{+21.0}$ | $74.5^{+17.6}$ | $85.8^{+22.5}$ |
| Caltech-101 | 66.7 | 61.5 | 63.8 | $88.5^{+32.7}$ | $83.1^{+35.1}$ | $86.1^{+34.9}$ | $91.9^{+37.8}$ | $83.6^{+35.9}$ | $87.8^{+37.7}$ |
| Flowers | 78.0 | 72.6 | 75.2 | $93.7^{+20.1}$ | $90.1^{+24.1}$ | $92.7^{+23.3}$ | $93.7^{+20.1}$ | $86.1^{+18.5}$ | $92.4^{+22.9}$ |

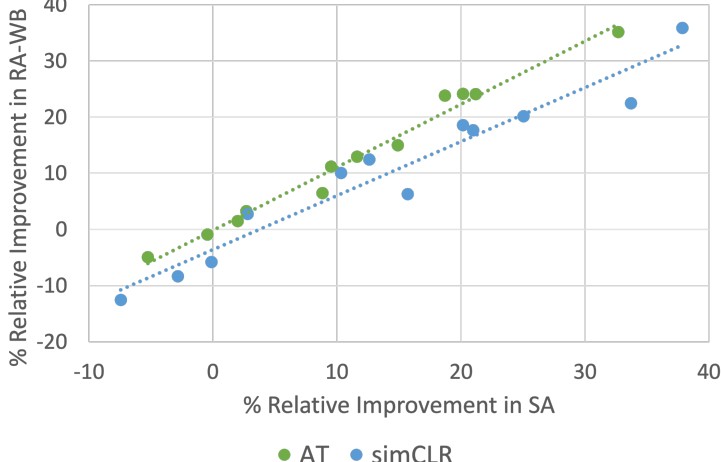

Figure 2: Plotting the improvement (%) introduced by pre-training relative to random initialization across all 12 target tasks. Improvement in RA-WB is linearly correlated with improvement in SA for both pre-training methods, with $R^2$ value of 0.98 for AT and 0.94.

increase in SA due to pre-training. We observe a strong linear correlation between the two quantities for both the pre-training methods, with $R^2$ value of 0.98 for AT and 0.94 for SimCLR.

## 4.3 Certified Adversarial Robustness

To the best of our knowledge, there exist no works that explicitly study the utility of transfer learning in the context of certified adversarial robustness for either supervised or self-supervised pre-training. As before, we train three versions of a ResNet-50 classifier on each target task: (i) using randomly initialized weights, (ii) using pre-trained weights obtained by performing Consistency Regularization (CR) (Jeong & Shin, 2020)

Table 5: Evaluating the benefits of pre-training on certified adversarial robustness. Given a target task, we train three ResNet-50 classifiers: one using random weight initialization and two using weights pre-trained on a source task (ImageNet). Pre-training is performed using supervised (consistency regularization) and self-supervised (SimCLR) objectives. During fine-tuning, the full network is trained using CR. In all three cases, training on target tasks is performed using consistency regularization. Similar to empirical adversarial robustness, pre-training improves certified adversarial robustness across all target tasks. Relative change due to pre-training is denoted in superscript.

| | Random Init. | | | Sup. Pre-Training | | | Self-Sup. Pre-Training | | |
|---|---|---|---|---|---|---|---|---|---|
| Target Task | SA (%) | RA-CT (%) | ACR ($\ell_2$) | SA (%) | RA-CT (%) | ACR ($\ell_2$) | SA (%) | RA-CT (%) | ACR ($\ell_2$) |
| Food | 63.0 | 53.9 | 0.891 | $63.2^{+00.3}$ | $53.5^{-00.7}$ | $0.874^{-01.9}$ | $64.4^{+02.2}$ | $57.6^{+07.0}$ | $0.923^{+03.5}$ |
| CIFAR-100 | 70.0 | 62.8 | 1.075 | $70.8^{+01.1}$ | $65.2^{+03.8}$ | $1.101^{+02.4}$ | $72.8^{+04.0}$ | $65.0^{+03.5}$ | $1.089^{+01.3}$ |
| CIFAR-10 | 89.6 | 86.0 | 1.508 | $93.4^{+04.2}$ | $89.2^{+03.7}$ | $1.601^{+06.1}$ | $93.2^{+04.0}$ | $90.4^{+05.1}$ | $1.619^{+07.4}$ |
| Birdsnap | 42.0 | 34.7 | 0.538 | $41.6^{-00.9}$ | $32.4^{-06.6}$ | $0.504^{-06.3}$ | $41.0^{-02.3}$ | $35.3^{+01.7}$ | $0.541^{+00.5}$ |
| SUN397 | 37.0 | 32.5 | 0.519 | $42.3^{+14.4}$ | $37.4^{+15.1}$ | $0.586^{+13.0}$ | $44.1^{+19.1}$ | $36.2^{+11.5}$ | $0.585^{+12.7}$ |
| Caltech-256 | 54.0 | 47.4 | 0.835 | $60.9^{+12.8}$ | $57.3^{+20.8}$ | $1.001^{+19.9}$ | $65.4^{+21.2}$ | $58.5^{+23.3}$ | $1.000^{+19.7}$ |
| Cars | 81.9 | 77.5 | 1.358 | $79.1^{-03.4}$ | $73.9^{-04.6}$ | $1.285^{-05.4}$ | $77.7^{-05.1}$ | $70.1^{-09.5}$ | $1.158^{-14.7}$ |
| Aircraft | 70.1 | 63.4 | 1.065 | $68.1^{-02.8}$ | $60.9^{-04.0}$ | $1.022^{-04.0}$ | $69.0^{-01.5}$ | $61.4^{-03.2}$ | $0.991^{-06.9}$ |
| DTD | 44.9 | 39.4 | 0.699 | $50.2^{+11.8}$ | $45.3^{+15.1}$ | $0.790^{+13.0}$ | $55.5^{+23.7}$ | $49.8^{+26.5}$ | $0.849^{+21.5}$ |
| Pets | 66.7 | 61.8 | 1.068 | $70.8^{+06.2}$ | $64.5^{+04.4}$ | $1.088^{+01.9}$ | $75.2^{+12.8}$ | $67.2^{+08.7}$ | $1.089^{+02.0}$ |
| Caltech-101 | 62.8 | 58.3 | 1.019 | $78.6^{+25.2}$ | $76.0^{+30.4}$ | $1.339^{+31.4}$ | $80.3^{+28.0}$ | $73.7^{+26.4}$ | $1.300^{+27.5}$ |
| Flowers | 75.2 | 72.7 | 1.306 | $87.5^{+16.4}$ | $82.0^{+12.9}$ | $1.538^{+17.7}$ | $84.6^{+12.5}$ | $78.7^{+08.3}$ | $1.407^{+07.7}$ |

\* The above results are generated by evaluating a smooth classifier. This entails performing the computationally expensive process of certification, which scales poorly with input dimension. Since all our datasets are ImageNet size (*i.e.,* $224 \times 224$), we follow the standard practice (Cohen et al., 2019) and perform certification using only 500 evenly spaced images in the test set.

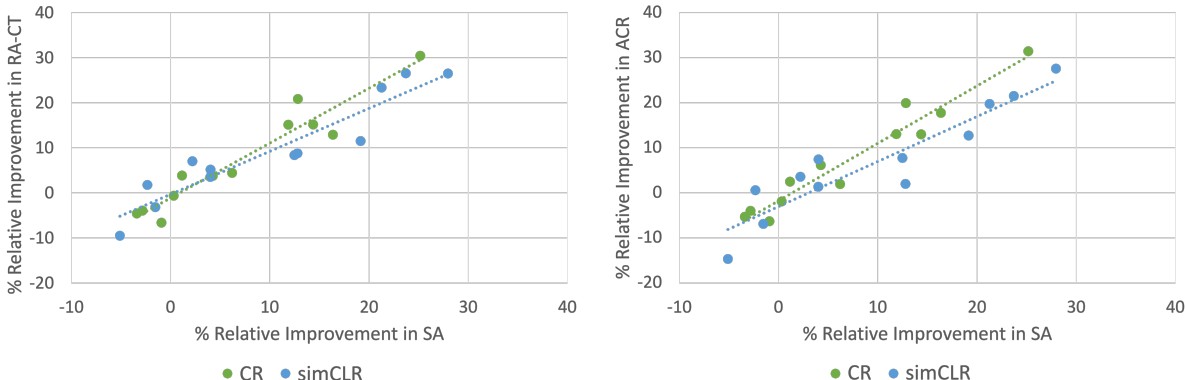

Figure 3: Plotting the improvement (%) introduced by pre-training relative to random initialization across all 12 target tasks. Improvements in both RA-CT (left) and ACR (right) are linearly correlated with improvement in SA for both pre-training methods. For the left plot, $R^2$ values for CR and SimCLR are 0.92 and 0.89. For the right plot, $R^2$ values are 0.94 and 0.86.

on ImageNet, and (iii) using pre-trained weights obtained by performing SimCLR on ImageNet. In order to achieve certified robustness during inference, we convert the ResNet-50 classifiers into smooth classifiers following Equation 1. The standard accuracy (SA), certified robust accuracy (RA-CT), and Average Certified Radius (ACR) of the smooth classifiers are reported in Table 5. We compute these quantities using the prediction and certification process described by Cohen et al. (2019).

We observe that supervised and self-supervised pre-training improves performance (SA) and certified robustness (RA-CT and ACR) on downstream tasks. Pre-training with CR results in an average relative improvement of 7.1%, 7.5%, and 7.3% compared to no pre-training on SA, RA-CT, and ACR, respectively. Similarly, Pre-training with SimCLR results in an average relative improvement of 9.9%, 9.1%, and 6.9% compared to no pre-training on SA, RA-CT, and ACR, respectively. As before, we note that the improvements in RA-CT and ACR are not necessarily due to the "transfer" of robustness of the pre-trained model. Rather, the improvement in SA seems to result in an overall increase in RA-CT and ACR. In Figure 3, we plot both the relative improvement in SA vs. RA-CT and SA vs. ACR from pre-training and see a strong linear correlation between these quantities. For CR pre-training, the $R^2$ value for linear correlation between SA and RA-CT is 0.92, and between SA and ACR is 0.94. For SimCLR pre-training, the $R^2$ value for linear correlation between SA and RA-CT is 0.89, and between SA and ACR is 0.86.

## 5  Discussion

In Section 4, we demonstrated that a robust transfer learning pipeline is an effective method to train robust models, especially on downstream tasks with small amounts of labeled data. In fact, our self-supervised pre-training results highlight that a large **labeled** pre-training dataset is also unnecessary. However, there remains a question as to which parts of the robust transfer learning pipeline need to use robust training methods. As robust training methods impose a higher training overhead compared to non-robust training methods (Shafahi et al., 2019a; Wong et al., 2019; Vaishnavi et al., 2022), we perform two additional experiments to understand which parts of the transfer learning pipeline must use robust training methods.

### 5.1  Is Robust Pre-training Necessary?

Transfer learning is designed to improve standard performance on downstream tasks. In Section 4, we observed a strong linear correlation between improvements in performance *vs.* improvements in robustness. This observation suggests that the robustness of the pre-trained model may be irrelevant. The SimCLR results provide further evidence as this training method does not optimize for robustness, and the models trained with it possess no empirically or certifiable robustness. Using the same experimental setup as in Section 4, we pre-train a ResNet-50 model using standard training (**ST**), *i.e.,* minimizing the cross entropy loss, which is also a non-robust pre-training method like SimCLR. We still perform robust fine-tuning of the full network. In Table 6, we measure the empirical and certified robustness of models pre-trained with ST on two downstream datasets and compare it to pre-training with SimCLR and the respective robust pre-training method. We only see minor differences when using ST and SimCLR compared to a robust pre-training method, suggesting that robust pre-training is unnecessary for improving robustness on the downstream task.

To further corroborate our claim that robustness on source task is unnecessary when looking to improve robustness on downstream target tasks, we refer to the prior work by Yamada & Otani (2022). In the context of robustness against Gaussian noise corruption, Yamada *et al.* state that *"it seems difficult to transfer corruption robustness from ImageNet to CIFAR-10. In fact, we find that a non-robustified ImageNet pre-trained ResNet-50 performs the best when fine-tuned for CIFAR-10"*. Their findings suggest that Gaussian noise robustness on downstream tasks (CIFAR-10) does not benefit from Gaussian noise robustness on the source task (ImageNet), implying that corruption robustness does not transfer between tasks. We observe a similar result but in the context of adversarial noise instead of Gaussian noise.

### 5.2  Is Robust Fine-tuning Necessary?

In our initial experiments with a robust pre-trained model, we found that we could not use standard training and fine-tune the entire model. The resulting model exhibited neither empirical nor certified robustness as it was biased towards maximizing standard performance. However, Shafahi et al. (2019b) showed that it was possible to train an empirically robust network (against white-box and black-box attacks) if standard fine-tuning was only done on the last model layer, thus freezing the rest of the model, which was pre-trained using AT. The intuition is that the frozen layer of the model pre-trained with AT acts as a robust feature

Table 6: Effect of the pre-training method on empirical and certified robustness. The full network is fine-tuned using AT and CR, respectively. Robustness is not a requirement during pre-training in order to observe improvement in robustness on downstream tasks. Change due to using non-robust pre-training methods (ST, simCLR) relative to using robust pre-training (AT/CR) is denoted in superscript.

| Task | Empirical Robustness | | | | Certified Robustness | | | |
|------|-------------|-----------|------------|------------|-------------|-----------|-----------|------------|
| | Pre-Training | SA (%) | RA-WB (%) | RA-BB (%) | Pre-Training | SA (%) | RA-CT (%) | ACR ($\ell_2$) |
| CIFAR-10 | AT | 95.8 | 91.7 | 94.0 | CR | 93.4 | 89.2 | 1.601 |
| | ST | $95.4^{-0.5}$ | $91.2^{-0.5}$ | $91.4^{-2.8}$ | ST | $93.0^{-0.4}$ | $88.6^{-0.7}$ | $1.584^{-1.0}$ |
| | SimCLR | $95.9^{+0.1}$ | $91.2^{-0.5}$ | $93.8^{-0.3}$ | SimCLR | $93.2^{-0.2}$ | $90.4^{+1.3}$ | $1.619^{+1.2}$ |
| CIFAR-100 | AT | 80.1 | 70.6 | 75.1 | CR | 70.8 | 65.2 | 1.101 |
| | ST | $78.5^{-2.1}$ | $68.1^{-3.5}$ | $73.5^{-2.2}$ | ST | $70.2^{-0.8}$ | $60.6^{-7.1}$ | $1.050^{-4.7}$ |
| | SimCLR | $80.9^{+0.9}$ | $70.3^{-0.5}$ | $75.8^{+0.9}$ | SimCLR | $72.8^{+2.8}$ | $65.0^{-0.3}$ | $1.089^{-1.1}$ |

Table 7: Studying whether certified robustness is preserved on fine-tuning the final layer of a pre-trained model non-robustly using standard training (*i.e., $\sigma = 0$*). Using different values of $\sigma$ during training and inference causes the smooth classifier to exhibit poor SA, RA, and ACR.

| Task | $\sigma = 0.5$ | | | $\sigma = 0.0$ | | |
|------|---------|-----------|------------|---------|-----------|------------|
| | SA (%) | RA-CT (%) | ACR ($\ell_2$) | SA (%) | RA-CT (%) | ACR ($\ell_2$) |
| CIFAR-10 | 8.4 | 5.4 | 0.073 | 91.0 | 0.0 | 0.000 |
| CIFAR-100 | 0.4 | 0.4 | 0.008 | 75.6 | 0.0 | 0.000 |

extractor that can be fine-tuned non-robustly while preserving empirical robustness. Their method results in a less robust model compared to robust fine-tuning but is computationally more efficient. To verify whether their findings extend to certified robustness, we replicate their experiments by first pre-training a ResNet-50 network on ImageNet using Consistency Regularization (CR) with $\sigma = 0.5$ and then fine-tuning the final layer only on CIFAR-10 and CIFAR-100 using Standard Training. During inference, we convert the ResNet-50 classifier into a smooth classifier (with $\sigma = 0.5$) following Equation 1 to measure certified robustness.

In Table 7, we report the performance and robustness of our ResNet-50 classifiers when converted in a smooth classifier with $\sigma = 0.5$. We observe that on both datasets, non-robust fine-tuning of the last layer results in a classifier with trivial standard accuracy (SA), certified robust accuracy (RA-CT), and average certified radius (ACR). Recall from Equation 1 that a smooth classifier $g_\theta$ performs prediction by taking majority voting over several copies of a given input $x$ sampled from the distribution $\mathcal{N}(x, \sigma^2 I)$. Thus, the base classifier should be trained using a noisy distribution (*i.e., $\sigma = 0.5$*). Standard fine-tuning is equivalent to training with $\sigma = 0$. Thus, the smooth classifier's performance suffers. We see that if we instead use $\sigma = 0$, the SA of the smooth classifier is restored, though it has zero RA-CT and ACR (follows directly from Equation 2). From these results, we conclude that robust fine-tuning is a necessary step for robust transfer learning to avoid catastrophic forgetting of robustness on the downstream task. Although Shafahi et al. (2019b) demonstrate a potential alternative for this finding in the context of empirical robustness, it significantly lowers the performance and robustness of the fine-tuned model, and as we demonstrated, it doesn't extend to certified robustness.

# 6   Limitations

In this paper, we present a comprehensive analysis of the benefits of transfer learning in the context of adversarial robustness. Our experiments utilize a suite of 12 downstream datasets of varying sizes and several training methods that optimize a diverse set of objectives (pertinent to our study), including standard generalization, empirical robustness, and certified robustness. Through these experiments, we address the knowledge gap highlighted in Table 1 and offer a novel insight into why transfer learning enhances adversarial robustness. Despite covering many significant aspects of the experiment space, there are a few notable areas we have not addressed.

Firstly, all our experiments were conducted using the ResNet-50 classifier. We chose ResNet-50 as the prior works we relied on primarily focused on this model, making it easier to identify hyperparameters and access pre-trained models. However, the current state-of-the-art in image classification is transformer-based models (Dosovitskiy et al., 2020; Liu et al., 2021), not CNN-based models. Therefore, it is important to verify our findings using transformer-based models. Secondly, we only used a contrastive self-supervised learning method in our experiments. As non-contrastive self-supervised learning methods (Grill et al., 2020; Chen & He, 2021; He et al., 2022) gain traction, future works should analyze these methods within our setup. Thirdly, as mentioned in Section 4.1, we limited our experiments to the $\ell_2$ threat model to ensure comparability of results in the empirical and certified robustness sections. However, prior works on empirical robustness primarily focus on the $\ell_\infty$ threat model. Lastly, we only provide empirical evidence for our claim that the effectiveness of transfer learning in improving adversarial robustness is due to an increase in standard accuracy, not the direct "transfer" of robustness from the source to the target task. Theoretical evidence is needed to further solidify this claim.

# 7   Conclusion

In summary, our research demonstrates that transfer learning, typically employed to enhance classifiers' standard generalization on tasks with limited labeled training data, can effectively contribute to improving adversarial robustness on downstream tasks, even when the sample complexity of robust generalization is significantly higher than that of standard generalization. This result holds irrespective of the amount of training data available for the downstream task.

Our experiments reveal that utilizing non-robust training methods during pre-training can still yield benefits for adversarial robustness on downstream tasks, provided robust training methods are employed during fine-tuning. We show for the first time that, contrary to traditional beliefs, the gains in robustness on the downstream task can be attributed to an increase in standard accuracy—a byproduct of transfer learning—rather than a direct "transfer" of robustness from the source to the target task.

We also find that pre-training can be performed using unlabeled data only by leveraging self-supervised training methods. Across 12 downstream tasks, employing (non-robust) self-supervised pre-training on ImageNet enhances average empirical and certified robustness by 9.9% and 6.9%, respectively. Our work stands as the first demonstration of training certifiably robust classifiers on tasks with extremely limited labeled training data.

## Acknowledgement

This work is supported in part by the Air Force Research Lab under grants FA9550-22-1-0029 and FA9550-22-1-0450 and Office of Naval Research award N00014-22-1-2001. Any opinions, findings, or conclusions expressed in this material are those of the authors and do not necessarily reflect the views of the sponsors.

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

# A    Implementation Details

To promote reproducibility, we provide all necessary implementation details in this Appendix. Statistics regarding all the datasets we used in our experiments are provided in Table 8. To train deep neural networks, we use the open-source PyTorch library (Paszke et al., 2019). For adversarial training, we use the open-source Robustness library (Engstrom et al., 2019) developed by Madry Lab. For autoPGD attack evaluation, we use the AutoAttack official code.[1] For training and evaluation using the randomized smoothing framework, we use the code provided by Jeong *et al.* (Jeong & Shin, 2020).[2] All the code used to produce the results in this paper is available at https://github.com/Ethos-lab/transfer_learning_for_adversarial_robustness.

Table 8: Statistics for all datasets used in our experiments.

| Dataset | # Train Images | # Classes | # Test Images | Skip | # Certified Images |
|---|---|---|---|---|---|
| ImageNet | 1,281,167 | 1,000 | 50,000 | 100 | 500 |
| Food | 75,750 | 101 | 25,250 | 50 | 505 |
| CIFAR-10/100 | 50,000 | 10/100 | 10,000 | 20 | 500 |
| Birdsnap | 32,677 | 500 | 8,171 | 16 | 511 |
| ☼ | 19,850 | 397 | 19,850 | 39 | 509 |
| Caltech-256 | 15,420 | 257 | 15,189 | 30 | 506 |
| Cars | 8,141 | 196 | 8,041 | 16 | 503 |
| Aircraft | 6,667 | 100 | 3,333 | 6 | 556 |
| DTD | 3,760 | 47 | 1,880 | 4 | 470 |
| Pets | 3,680 | 37 | 3,669 | 7 | 524 |
| Caltech-101 | 3,030 | 101 | 5,647 | 11 | 513 |
| Flowers | 2,040 | 102 | 6,149 | 12 | 512 |

***Input Pre-processing.*** For all experiments, we fix the dimension of the input image to $224 \times 224$. For cases where the image is of smaller resolution (*i.e.,* CIFAR-10 and CIFAR-100), we upscale it first during the input pre-processing stage. The complete set of pre-processing steps we perform are as follows:

```
TRAIN_TRANSFORMS = transforms.Compose([
    transforms.RandomSizedCrop(224),
    transforms.RandomHorizontalFlip(),
    transforms.ToTensor(),
    transforms.Normalize()
])

TEST_TRANSFORMS = transforms.Compose([
    transforms.Scale(256),
    transforms.CenterCrop(224),
    transforms.ToTensor(),
    transforms.Normalize()
])
```

We follow prior works (He et al., 2019; Kornblith et al., 2019; Salman et al., 2020) and only use normalization for the ImageNet, CIFAR-10, and CIFAR-100 datasets.

***Training.*** When training from scratch, we perform hyperparameter tuning by performing grid search over lr $\in \{0.1, 0.01, 0.05, 0.001\}$, batch size $\in \{256, 128, 64, 32\}$, and weight decay $\in \{1e-04, 1e-03, 1e-02\}$.

---

[1] https://github.com/fra31/auto-attack
[2] https://github.com/jh-jeong/smoothing-consistency

Before terminating training, the learning rate is decayed twice by a factor of 0.1 when the performance on validation set doesn't improve for 30 epochs. For ImageNet pre-training, we use publicly available weights for Adversarial Training[3] and SimCLR.[4] Since ImageNet pre-trained weights are not publicly available for the Consistency Regularization method, we generate them ourselves using hyperparameter details provided by the authors (Jeong & Shin, 2020). For all training, we use the Stochastic Gradient Descent (SGD) optimizer.

***Certification Using Randomized Smoothing.*** During certification, we use $\sigma = 0.5$ and follow Cohen *et al.* (Cohen et al., 2019) for all other hyperparameters, *i.e.,* $N_0 = 100$, $N = 100,000$, and failure probability $\alpha = 0.001$. Also following prior works, we certify about 500 test images for each dataset, by skipping every $n^{th}$ image in the complete test set (see Table 8 for skip factor used).

---

[3] https://github.com/microsoft/robust-models-transfer
[4] https://github.com/facebookresearch/vissl/blob/main/MODEL_ZOO.md

