# OpenReview forum: "A Study of the Effects of Transfer Learning on Adversarial Robustness"
_TMLR — Accepted by TMLR_

### Review · Reviewer_HEfa · 2024-02-09

**Summary Of Contributions:**

This paper proposes conducted an empirical studies on the empirical and certified robustness of a robustly pre-trained model after robust fine-tuning. They are for the first time to evaluate the effect of transfer learning on certified accuracy. Their findings indicate that robust pre-training is unnecessary while robust fine-tunning is necessary.

**Audience:**

Yes

**Claims And Evidence:**

Yes

**Requested Changes:**

Please see the weaknesses above.

**Strengths And Weaknesses:**

Strengths:

1. The authors make the first effort to study the certified robustness achieved by robust fine-tuning.

Weaknesses:

1. Section 3 seems to be redundant. The conclusion that ‘Empirical is not the same as Certified’ is not surprising and irrelevant to main topic of this paper.

2. The empirical conclusion could be wrong. The authors falsely take pre-trained ResNet-50 via SimCLR as the robustly pre-trained models. Therefore, the obtained conclusions and analyses are not reliable. For example, the conclusion that robust pre-training is unnecessary could be incorrect according to that pre-trained models via standard training and SimCLR achieves similar robustness.

3. The critical claim that improvements in RA due to transfer learning are largely due to the overall improvement in SA seems to be meaningless. According to TRADES, the robust error can be decomposed into the sum of standard error and boundary error. If standard error decreases, that the robust error should correspondingly decrease. Therefore, I am confused about how such claim can provide some insights for conducting robust fine-tuning.

4. The authors does not cite adequate related works, including [1,2] in the field of robust self-supervised learning and [3,4] in the field of robust fine-tuning. Besides, the authors should utilize the state-of-the-art robust fine-tuning methods [3,4] to conduct the experiments.

[1] Efficient Adversarial Contrastive Learning via Robustness-Aware Coreset Selection, Xu et al., NeurIPS 2023. \
[2] Enhancing Adversarial Contrastive Learning via Adversarial Invariant Regularization, Xu et al., NeurIPS 2023. \
[3] TWINS: A Fine-Tuning Framework for Improved Transferability of Adversarial Robustness and Generalization, CVPR 2023. \
[4] AutoLoRa: A Parameter-Free Automated Robust Fine-Tuning, Xu et al., ICLR 2024.

5. The authors only conducted the experiments on ResNet models. I suggest studying the robustness of pre-trained ViT after robust fine-tuning.

---

### Review · Reviewer_4th7 · 2024-02-16

**Summary Of Contributions:**

This article focuses on robustness in the field of transfer learning. By experimenting with robust fine-tuning of models with random initialization, adversarial pre-training, and self-supervised pre-training on many downstream tasks, the authors find that improving the standard accuracy of the pre-trained model is a key point to improving downstream robustness. The authors then explore the roles of robust pre-training and robust fine-tuning and suggest that robust fine-tuning is necessary for obtaining robustness.

**Audience:**

Yes

**Broader Impact Concerns:**

No particular concerns have been identified.

**Claims And Evidence:**

No

**Requested Changes:**

1. The PGD/AutoPGD attacks used for AT and robustness evaluation may not be an ideal choice since the focus of this paper is L2 robustness. Moreover, the robustness should be evaluated against the full AutoAttack, as otherwise can be taken as reliable robustness.

2. Transformer models should also be tested to obtain more general claims, as they are the mainstream pre-trained vision models now.

**Strengths And Weaknesses:**

Strengths:
1. The problem studied in this article is meaningful because the application scenarios of transfer learning are vast and the robustness of this is worth studying. The authors consider both empirical robustness and verifiable robustness, making the study comprehensive. Meanwhile, the paper is well-written and easy to follow.

2. In this paper, an interesting and plausible argument is made that the improvement of clean accuracy of pre-trained models is the key to improving the downstream model’s robustness, and it is validated with sufficient experiments.

Weaknesses:
1. The discussion of self-supervised pre-training would be better improved with the addition of a discussion of some kind of Masked Image Modeling methods since this training method differs from models trained by Contrastive Learning in terms of robustness.

2. Section 5.1 discusses robust pretraining and concludes robust pre-training is unnecessary for improving robustness on the downstream task, this sounds like it negates the meaning of robust pre-training. A more detailed discussion is possible here. although robust pre-training with normal fine-tuning does not increase white-box robustness, does it increase robustness against black-box attacks?

---

> ### Author Response · Authors · 2024-03-01
> **Rebuttal response for Reviewer 4th7**
>
> We thank the reviewer for their insightful review. We have identified two major issues highlighted by the reviewer, and we address and provide clarification on each of them.
>
> **Does robust pre-training with normal fine-tuning improve robustness against black-box attacks? + Evaluation using AutoAttack.**
>
> In [a], authors comment that *“It might be natural to expect that in the regime of infinite data, the Bayes-optimal classifier—the classifier minimizing classification error with full-information about the distribution—is a robust classifier. But this is not true.”* This comment relates to their key finding that training objectives associated with standard generalization and adversarial robustness are distinct and both need to be optimized jointly during training to simultaneously achieve high performance and robustness. Our experimental results are consistent with the findings of [a]: full network fine-tuning using only a non-robust objective results in the network achieving high non-robust performance, but causes it to **catastrophically forget all the robustness** (white-box and black-box). Even when using robust pre-trained weights, this robustness **does not** carry forward to the downstream task when full network fine-tuning is performed non-robustly. This catastrophic forgetting of robustness was also observed by [b] and [c]. As such, no matter what pre-trained weights we use, we **do not** expect any robustness to be preserved if full network fine-tuning is done non-robustly. We will add this discussion to the paper and back it up empirically using Square Attack (the black-box attack from the AutoAttack ensemble).
>
> In this paper, we examine the white-box robustness of networks in the l2-space (why we chose l2-space is explained in section 4.1). We found that state-of-the-art papers that publish results in the l2-space use AutoAttack (AA) in their evaluations [d]. We are unable to run the entire AA (ensemble of 4 attacks) as it is computationally very expensive (~ 35 hrs per model using our machine), and evaluating all our models (12 x 3 x 2 = 72) using AA would take about 4 months. Instead, we resorted to using autoPGD for three main reasons (see Table 2 in [e]): (i) it appears to be the strongest white-box attack within the AA ensemble, (ii) there is only a minor difference (~ 0.71%) between adversarial accuracies computed using autoPGD-CE and the full AA against an adversarially trained model, and (iii) autoPGD runs significantly faster (~ 8 hrs per model) than AA.
>
> We propose an alternative to running the full AA evaluation that takes into account our compute restrictions as well as the necessity to comprehensively evaluate the robustness of our networks, i.e., adding Square Attack results to the paper. In combination with the autoPGD results we already have, this should provide an accurate measurement of the networks’ robustness while allowing us to generate all results in a timely fashion.
>
> [a] Robustness May Be at Odds with Accuracy, ICLR 2019
>
> [b] Adversarially Robust Transfer Learning, ICLR 2020
>
> [c] Does Robustness on ImageNet Transfer to Downstream Tasks?, CVPR 2022
>
> [d] https://robustbench.github.io/
>
> [e] Reliable Evaluation of Adversarial Robustness with an Ensemble of Diverse Parameter-free Attacks, ICML 2020
>
> **Including Masked Image Modeling and transformer-based classifiers.**
>
> We have used 4 different pre-training methods in our paper: adversarial training, consistency regularization, SimCLR, and standard training. Collectively, these cover a wide-space of pre-training strategies, including different training objectives (non-robust/robust), different types of robustness (empirical/certified), and different learning methods (supervised/self-supervised). Our findings remain consistent across all these pre-training methods, which lends credibility to the generalization of these findings. However, if the reviewer strongly believes that adding a new type of self-supervised method will significantly improve the credibility of generalization of our results, we will include these results in the final version of the paper.
>
> We will include results using ViT in the final version of the paper to demonstrate the generalization of our findings across distinct architecture families.

---

### Review · Reviewer_rRES · 2024-02-25

**Summary Of Contributions:**

This paper presents an empirical study on the effectiveness of transfer learning for adversarial robustness.
In particular, the paper aims to demonstrate whether transfer learning can be adapted to transform adversarial robustness on a source dataset (ImageNet in this case) to other down-stream tasks.
To this end, the paper targets adversarial training and certified robustness in terms of $\ell_2$ norm as its target tasks.
Then, for each case, it studies the empirical differences between three models: i) models trained directly on the down-stream task, ii) using pre-trained weights of a classifier trained with either adversarial training or certified robustness and fine-tuning on the down-stream task, or iii) using weights of a pre-trained model using self-supervised training.
The empirical studies on various datasets with different sizes indicate that transfer learning is an effective way of improving the natural and robust accuracy.
Also, the findings suggest that the increase in robust accuracy might be the result of an increase in the natural accuracy.

**Audience:**

Yes

**Claims And Evidence:**

Yes

**Requested Changes:**

- Extending experimental settings as explained.
- Explaining the importance of the current work and how the community can benefit from this research.

**Strengths And Weaknesses:**

### Strengths:
- The paper is easy to read and follow.
- The experimental results suggest that using transfer learning can be an effective way of fine-tuning robust neural networks on down-stream tasks even in low-data regimes.
- The findings not only apply to adversarial robustness, but they also hold for certified robustness.

### Weaknesses:
- Even though the empirical observations are appreciated, I am not sure to what extent this paper provides novel AND helpful information. In other words, since the community already knew that transfer learning would work for rich-data regimes, this paper only extends those findings to other low-data regime datasets. But at the core, they both say that transfer learning is helpful for adversarial robustness.
- The experimental studies are a bit too restricted to make a general conclusion. The least the I expect is to use several architectures (such as WideResNet, DenseNet, ViTs, etc.), various $\ell_p$ norms, and various adversarial perturbation levels. With the current setting (look at table 6) it feels that the selected $\ell_2$ norm is too restrictive that transfer learning is just done for a regular vanilla task that is not intended to be robust.

---

> ### Author Response · Authors · 2024-03-01
> **Rebuttal response for Reviewer rRES**
>
> We are grateful for the reviewer's insightful feedback. We address and provide clarification on each of the raised issues below:
>
> ## Weakness 1: Novelty
>
> The main contributions of the paper can be summarized as follows:
>
> **1. Validating findings in low-data regime for empirical robustness**
>
> Within empirical adversarial robustness, we demonstrate that transfer learning can be beneficial in the low-data regime. Even though prior works showed this for high-data regimes, other prior works have also shown that the benefits of transfer learning (for standard generalization) can be **inconsistent** between high and low-data regimes [a]. Therefore our results ascertain that this inconsistency **doesn’t exist** when training empirically robust models.
>
> [a] Rethinking ImageNet Pre-training, ICCV 2019
>
> **2. Validating findings in high and low-data regime for certified robustness**
>
> There are **no previous works** on whether transfer learning improves certified robustness, our work is the first to present these results. Due to the perceived similarities between empirical and certified robustness, one might feel compelled to translate the findings from one type of robustness to another. However, as discussed in Section 3, such a translation **does not** always hold and needs to be empirically validated to avoid ending up with erroneous conclusions.
>
> **3. Adding more nuance to the transfer learning framework**
>
> While traditional transfer learning concerns itself with a single objective only (i.e., performance), transfer learning for adversarial robustness needs to take into account two distinct objectives that are not complementary (i.e., performance and robustness) [b]. As such, when studying transfer learning in the context of adversarial robustness, one needs to ascertain the value of each objective during different phases of transfer learning (i.e., pre-training and fine-tuning). We perform this analysis (Section 5) and find that robust pre-training **does not** significantly improve the robustness of the final model, contrary to what one might expect. Instead, the robustness objective is only necessary during fine-tuning.
>
> [b] Robustness May Be at Odds with Accuracy, ICLR 2019
>
> ## Weakness 2: Additional experiments
>
> We agree that additional experiments can improve the paper, but we have limited capacity to run experiments due to restrictions in computational resources. As such, we would like to work with the reviewer to identify the set of new experiments that are critical to the paper.
>
> **Network architectures**
>
> We will add results for ViT as we agree it is important to showcase the generalization of our results across distinct NN architectures.
>
> **$\ell_p$ norms**
>
> We chose the $\ell_2$ norm because randomized smoothing (our choice of certified robustness method) is only defined in the $\ell_2$ space, and we wanted to maintain a consistent threat model across our paper to avoid confusion. Note that robustness in the $\ell_2$ space can be directly converted into robustness in $\ell_\infty$ space. Linear algebra tells us that the $\ell_2$-ball of radius $\sqrt{d}$ contains the $\ell_\infty$ unit ball in $R^d$. So a model robust against perturbation of $\ell_2$ norm $r$ is also robust against perturbation of $\ell_\infty$ norm $r / \sqrt{d}$. In fact, prior work has used this formula to compute $\ell_\infty$ robustness for their $\ell_2$ robust models. We can follow a similar approach to show that our results hold in the $\ell_\infty$ space.
>
> **Perturbation levels ($\epsilon$)**
>
> We are unsure about the reviewer’s expectations with regard to the different perturbation levels experiment. We do not have any reason to believe that our conclusions may change on varying perturbation levels.
>
> Our expectations are as follows:
>
> 1. If we vary $\epsilon$ during training and evaluation (from low to high), we expect robustness numbers to decrease monotonically. However, this drop will occur for all models alike, irrespective of whether pre-training was performed or not. As such, we still expect pre-training to result in higher robustness as compared to no pre-training for all values of $\epsilon$.
>
> 2. If we vary $\epsilon$ during evaluation only, from 0 to the value used during training, the robustness will monotonically decrease. For higher values of $\epsilon$, we expect to see no meaningful robustness. We expect this for all models, whether pre-training is performed or not.
>
> We request the reviewer to share any prior work that challenges our expectations and demonstrates that changing perturbation levels can result in inconsistencies in final results.

---

### Review · Reviewer_3Lh2 · 2024-02-26

**Summary Of Contributions:**

The authors present an evaluation of robust transfer learning from ImageNet to a series of downstream tasks, considering both empirical and certified robustness.

**Audience:**

Yes

**Broader Impact Concerns:**

No concerns.

**Claims And Evidence:**

No

**Requested Changes:**

In order for this paper to be insightful, I think that significant work would be needed in terms of improving experimental setup and deriving useful insights for the reader. I do not think this is within the scope of a revision.

**Strengths And Weaknesses:**

Strengths:
- Good overview of what is unexplored in the literature.
- Considering both empirical and certified robustness.

Weaknesses:
- I am not sure what section 3 is supposed to tell me. Assuming that AT is trained with L2 adversaries, the experiments just highlight the two different notions of robustness. It is inherently not comparable and I am not convinced that the presented results give any more insights into this. Also I do not see how these experiments motivate the work on transfer learning.
- Use AutoAttack [1] instead of PGD with only 3 steps; PGD wit only 3 steps is hardly comparable across methods and datasets and will have extremely high variance (which is not reported).
- Having AT as supervised pre-training but non-robust self-supervised pre-training makes the results not comparable.
- Tables are overall a bit overwhelming; it would be more useful to highlight improvements and reductions in performance somehow and then discuss these cases in the text appropriately.
- Interesting cases where performance actually reduces (e.g., Cars, Aircraft) are not discussed.
- There are not insights provided into why there is no robustness transfer and improvements seem to come purely from standard performance improvements. That is not surprising for standard (non-robust) self-supervise pretraining, but for robust supervised pre-training this is surprising. But it is not further discussed or investigated.

Conclusion:
I am not convinced that this paper is a meaningful contribution for TMLR. I feel the experimental setup is not appropriate and insights are missing.

---

> ### Author Response · Authors · 2024-03-01
> **Rebuttal response for reviewer 3Lh2**
>
> We are grateful for the reviewer's insightful feedback. We address and provide clarification on each of the raised issues below:
>
> **Regarding Section 3**
>
> Prior reviews received for this paper indicated that reviewers tended to trivially translate the findings for empirical robustness to certified robustness and, as such, considered our results on certified robustness to be 'expected'. We revised our paper and added Section 3 to highlight that this expectation can be erroneous, and backed up this claim by presenting a couple of cases where such an expectation **doesn’t** hold. Through the discussion in Section 3, we motivate the need to empirically validate whether transfer learning benefits certified robustness similar to empirical robustness.
>
> **Attack used during evaluation**
>
> We use the PGD attack with 3 steps during adversarial training only. During evaluation, we use the autoPGD attack with 100 steps. We apologize for not including this detail in the paper and will add it moving forward.
>
> We chose autoPGD instead of the entire AutoAttack (AA) due to the following reasons:  (i) it appears to be the strongest white-box attack within the AA ensemble, (ii) there is only a minor difference (~ 0.71%) between adversarial accuracies computed using autoPGD-CE and the full AA against an adversarially trained model, and (iii) autoPGD runs significantly faster (~ 8 hrs per model) than AA (~ 35 hrs per model). We will add results for Square Attack (the attack most distinct to autoPGD) in the paper to make the results more convincing.
>
> **“Having AT as supervised pre-training but non-robust self-supervised pre-training makes the results not comparable”**
>
> In Tables 4/5, we train three different models for each of the 12 downstream tasks: (M1) starting with random weights, (M2) starting with AT/CR (on ImageNet) weights, and (M3) starting with simCLR (on ImageNet) weights. The fine-tuning is performed using AT/CR for all models. We first compare M1 with M2 and M1 with M3 separately to conclude that pre-training benefits adversarial robustness irrespective of the type of pre-training method used. Here, we purposefully select the **most diverse set** of pre-training methods to ensure the generalization of our results. Additionally, we observe that M2 and M3 models exhibit similar performance and robustness. This leads us to ask, *"is robustness a requirement during pre-training?"* and motivates Section 5.1, where we additionally compare supervised robust pre-training (AT/CR) with supervised non-robust pre-training (ST). Again we observe comparable results, allowing us to theorize that robustness is **not** a requirement during pre-training.
>
> **Regarding presentation of results table**
>
> We will follow the reviewer's suggestions and highlight improvements appropriately to make the tables easier to read.
>
> **Reason behind why robustness does not transfer**
>
> In Tables 4, 5, and 6, we observe that when using the same fine-tuning method, robust supervised pre-training (AT/CR) yields comparable results as non-robust supervised (ST) and non-robust self-supervised (simCLR) pre-training. This leads us to conclude that robustness is **not** being transferred between the source task and the target task. We refer to a prior work [a] to identify the possible reason behind this. In the context of robustness against Gaussian noise corruption, [a] find that *“it seems difficult to transfer corruption robustness from ImageNet to CIFAR10. In fact, we find that a non-robustified ImageNet pretrained ResNet performs the best when fine-tuned for CIFAR10.”* This suggests that Gaussian noise robustness on downstream tasks (CIFAR10) **doesn’t** benefit from Gaussian noise robustness on the source task (ImageNet), implying that robustness does not transfer. We observe a similar result but in the context of adversarial noise instead of Gaussian noise. We borrow the hypothesis of [a] that robust features overfit to the dataset and so are **not** as generalizable as features learned for standard generalization. We will add this discussion to the paper.
>
> [a] Does Robustness on ImageNet Transfer to Downstream Tasks?, CVPR 2022

---

### Decision · Action_Editor_4vpU · 2024-03-24

**Recommendation:** Accept with minor revision

**Comment:**

(The decision rationale was written by an EIC, following a discussion with AE and reading the reviews and author reponses).

Seeing that all reviewers answered positively as to whether the two TMLR criteria (claims matched by evidence and TMLR audience) were satisfied, it was determined that this submission should be accepted.

However, reviewers had some concerns, beyond the TMLR criteria, which we believe should be discussed in the final version of this submission, in what should be a minor revision.

Specifically, we ask the authors to add a new "Limitations" section to the submission, which explicitly highlights the weaknesses of the work described by the reviewers (such as the lack of experiments based on more recent architectures such as the ViT and MAE-based self-supervised learning, the reliance on white-box ℓ2 adversarial attacks only, the lack of a solid explanation for the empirical results reported beyond some intuitive hypotheses).

Additionally, please include any other changes / improvements mentioned in your author responses.

**Audience:**

Yes.

**Claims And Evidence:**

Reviewers have all acknowledged that claims made in the submission are supported by evidence, in their official recommendation.